# *Wolbachia* infection at least partially rescues the fertility and ovary defects of several new *Drosophila melanogaster bag of marbles* protein-coding mutants

**Miwa Wenzel**◉*, **Charles F. Aquadro***

Department of Molecular Biology and Genetics, Cornell University, Ithaca, New York, United States of America

* mjw349@cornell.edu (MW); cfa1@cornell.edu (CFA)

**Data Availability Statement:** All relevant data are within the manuscript and its Supporting information files.

## Abstract

The *D. melanogaster* protein coding gene *bag of marbles* (*bam*) plays a key role in early male and female reproduction by forming complexes with partner proteins to promote differentiation in gametogenesis. Like another germline gene, *Sex lethal*, *bam* genetically interacts with the endosymbiont *Wolbachia*, as *Wolbachia* rescues the reduced fertility of a *bam* hypomorphic mutant. Here, we explored the specificity of the *bam-Wolbachia* interaction by generating 22 new *bam* mutants, with ten mutants displaying fertility defects. Nine of these mutants trend towards rescue by the *w*Mel *Wolbachia* variant, with eight statistically significant at the fertility and/or cytological level. In some cases, fertility was increased a striking 20-fold. There is no specificity between the rescue and the known binding regions of *bam*, suggesting *w*Mel does not interact with one singular *bam* partner to rescue the reproductive phenotype. We further tested if *w*Mel interacts with *bam* in a non-specific way, by increasing *bam* transcript levels or acting upstream in germline stem cells. A fertility assessment of a *bam* RNAi knockdown mutant reveals that *w*Mel rescue is specific to functionally mutant *bam* alleles and we find no obvious evidence of *w*Mel interaction with germline stem cells in *bam* mutants.

## Author summary

Reproduction in the *Drosophila melanogaster* fruit fly is dependent on the *bag of marbles* (*bam*) gene, which acts early in the process of generating eggs and sperm. Mutations to this gene negatively impact the fertility of the fly, causing it to be sterile or have fewer progeny. Interestingly, we find that the bacteria *Wolbachia*, which resides within reproductive cells across a wide range of insects, partially restores the fertility and ovary phenotype of several *bam* mutants of which the resultant Bam protein is altered from wildtype. The protein function of Bam is further suggested to be important by the lack of rescue for a fly that has a fertility defect due to low expression of a non-mutated *bam* gene. Previous work makes similar conclusions about *Wolbachia* with another reproductive gene, *Sex*

**Funding:** This work was supported by the National Institute of Health grant R01-GM095783 to C.F.A. and by NYSTEM C029155 and NIH S10OD018516 grants to the Cornell Institute of Biotechnology Imaging Center for the Zeiss i880 LSM880 microscope. The funders had no role in study design, data collection and analysis, decision to publish, or preparation of the manuscript.

**Competing interests:** The authors have declared that no competing interests exist.

*lethal* (*Sxl*), highlighting the potential for rescue of fertility mutants to occur in a similar way across different genes. An understanding of the ways in which *Wolbachia* can affect host reproduction provides us with context with which to frame *Wolbachia*'s impact on host genes, such as *bam* and *Sxl*, and consider the evolutionary implications of *Wolbachia*'s infection in *D. melanogaster* fruit flies.

## Introduction

The *bag of marbles* (*bam*) gene plays a critical role in female and male gametogenesis in *D. melanogaster*. In the stem cell niche of the ovaries and testes lies a collection of germline stem cells (GSCs) that asymmetrically divide to self-renew and produce differentiating daughter cells. In females, *bam* acts as a master regulator and its expression induces daughter cell differentiation [1]. In males, *bam* functions later in gametogenesis where it initiates terminal differentiation of the spermatogonia that eventually become sperm [1–4].

In female *D. melanogaster*, *bam* is largely repressed in GSCs by signaling it receives from the stem cell niche [5–7]. Bam that escapes silencing in GSCs is antagonized by eukaryotic translation initiation factor 4A (eIF4A) [8]. When a GSC divides, a daughter cell is displaced from the stem cell niche, allowing *bam* to now be expressed. In the daughter cell, Bam functions to bring together various proteins that repress self-renewal and promote differentiation, leading to the establishment of a cystoblast.

In both male and female *D. melanogaster*, *bam* null mutants are completely sterile and have a tumorous cytological phenotype [1,9]. A partial loss-of-function mutant (*bam*^bw^) has reduced fertility in females and is sterile in males [10,11]. In females, the fertility defect of the *bam*^bw^ mutant is rescued by both *w*Mel and *w*MelCS variants of the endosymbiont bacteria *Wolbachia pipientis* [11,12].

*W. pipientis*, commonly referred to as *Wolbachia*, is estimated to inhabit 50% or more of insect species and is an obligate bacterium in filarial nematodes [13–15]. Across both arthropods and nematodes, *Wolbachia* manipulates reproduction to promote its own maternal transmission (reviewed in [16,17]). For example, *Wolbachia* famously induces cytoplasmic incompatibility, in which a female's progeny may have reduced viability if they do not harbor the same *Wolbachia* strain as the male parent. *Wolbachia* is also known to induce parthenogenesis, feminization, and male-killing of host progeny, all of which promote its maternal transmission. The extent of these parasitic phenotypes varies across *Wolbachia* variants and their host identity, age, and environment, e.g. [18–20]. Rescuing the fertility defect of the hypomorphic *bam* mutant aligns with the evolutionary interests of *Wolbachia*.

*Wolbachia* also rescues the sterility of hypomorphs of another GSC gene, *Sex lethal* (*Sxl*), in *D. melanogaster* (*Wolbachia* variant unspecified) [21]. It has thus been proposed that the rescue is dependent on the point mutation induced in the *Sxl* mutant and not on the severity of the mutant, though the researchers suggest that rescue cannot happen if *Sxl* protein levels are too low. Of note is that the *Sxl* mutants they studied were in different genetic backgrounds, which may have affected the results observed and the conclusions made. Such background effects were observed with *bam* in [22], where females with the same L255F *bam* mutation had strikingly different fertility across two different genetic backgrounds of *D. melanogaster*. Nevertheless, further research into one of the *Sxl* mutants, *Sxl*^f4^, showed that the rescue is partially due to a restoration of GSCs by the *Wolbachia* protein toxic manipulator of oogenesis (TomO) [23].

Mutants of additional germline genes *meiotic P26* (*mei-P26*), *ovarian tumor* (*otu*), and *sans fille* (*snf*) are not rescued by *Wolbachia*, despite having similar severity of an egg-absence phenotype to some *Sxl* mutants (*Wolbachia* variant unspecified) [21]. These results suggest the *Wolbachia* interaction may be specific to a functional pathway in female *D. melanogaster*, possibly through a direct, physical interaction with a gene product like Sxl that is expressed in the germline.

In this study, we explore the *bam-w*Mel interaction, building off the findings for the *bam^bw* hypomorphic mutant. The *bam^bw* hypomorphic allele is a L255F change in a gene region critical for *bam* function. Thus far, four functional regions of *bam* have been documented in *D. melanogaster*: The CAF40 binding motif (residues 13–36) [23], the Benign gonial cell neoplasm (Bgcn) binding region (residues 100–350) [24], the COP9 signalosome 4 (Csn4) binding region (residues 151–350) [24], and a ubiquitin binding region (residues 200–250) [25]. *Bam* also has a PEST domain (residues 402–434), which marks the protein for rapid degradation [1]. *Bam*'s specific binding regions and their interactions with binding partners are illustrated in Fig 1a and 1b respectively.

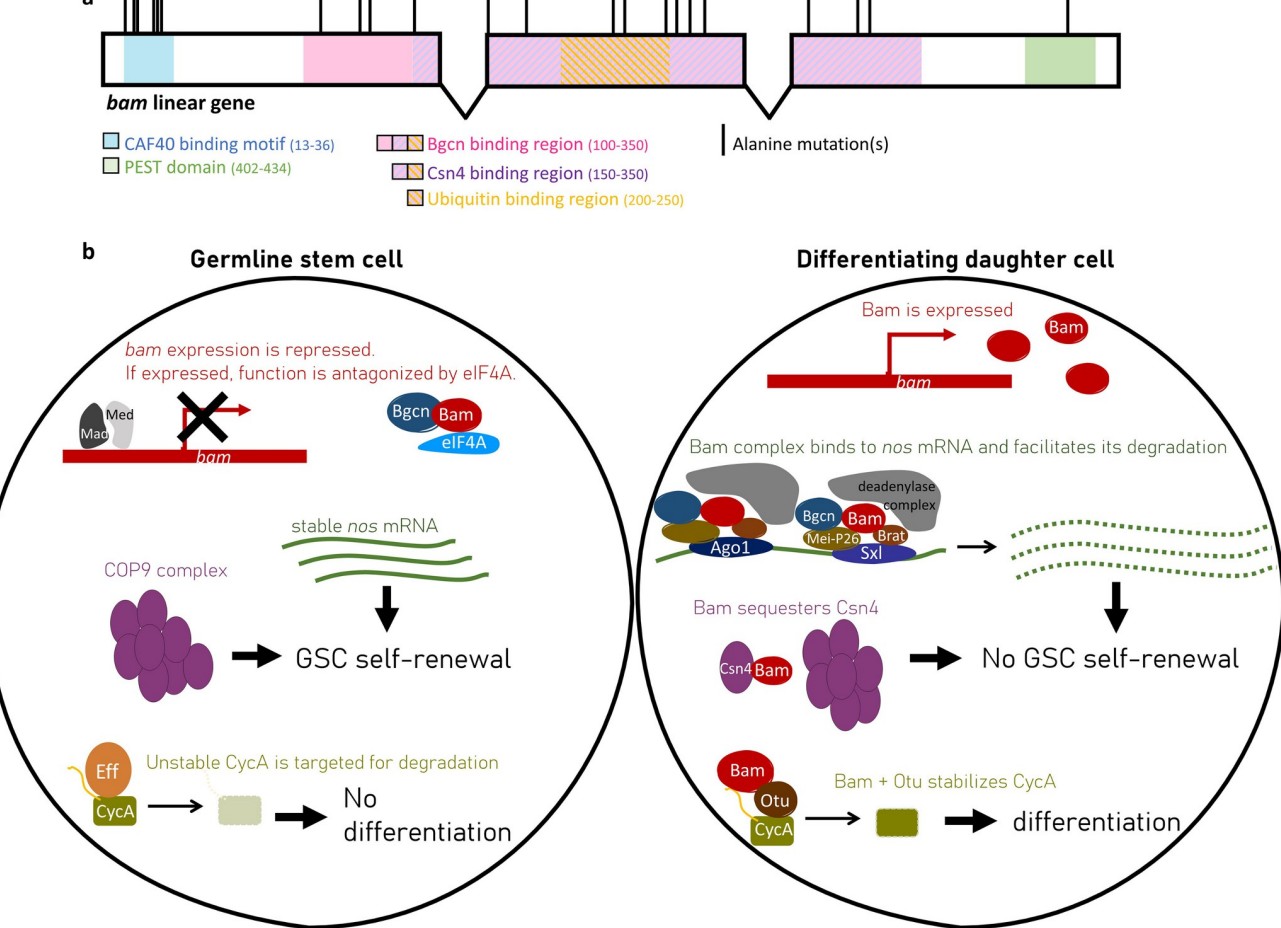

**Fig 1. *Bam* is involved in several processes that regulate the differentiation of GSCs in female *D. melanogaster*.** (a) Schematic of the five known binding or functional regions of *bam* are colored across the coding sequence exons of the gene. Each black vertical bar represents the location of a new mutant generated with alanine replacements in this study. (b) In GSCs (left), Bam is repressed and antagonized by eIF4A. In the differentiating daughter cells (right), Bam is expressed and coordinates other molecules to inhibit the self-renewal pathway or promote the differentiation pathway. Bam has been shown to genetically and physically interact with additional molecules that are not shown.

To repress GSC self-renewal, Bam forms a complex with Bgcn, Meiotic-P26 (Mei-P26), Brain tumor (Brat), a deadenylase, and either Argonaute-1 (Ago1) or Sxl to destabilize *nanos* (*nos*) mRNA [26–28]. The destabilization of *nos* mRNA may occur through Bam's interaction with the CAF40 which is part of the CCR4-NOT deadenylase complex [23,29]. Thus, in GSCs, *nos* mRNA is translated to promote GSC self-renewal; in differentiating daughter cells, *nos* mRNA is destabilized by the Bam complex, inhibiting *nos* translation and the self-renewal pathway.

Bam also represses the self-renewal pathway through its interaction with Csn4 [24]. In GSCs, the COP9 complex is composed of nine Csn subunits. When Bam is expressed in differentiating daughter cells, it breaks up the complex by sequestering Csn4, thereby inhibiting the GSC self-renewal pathway.

Additionally, Bam promotes differentiation by forming a complex with Otu to deubiquitinate Cyclin A (CycA) [25]. In GSCs, CycA is degraded through a process involving the ubiquitin-conjugating enzyme Effete [30]. In differentiating GSC daughter cells, it is proposed that Bam binds to both Otu and to proteins such as CycA, which have ubiquitin chains that mark them for degradation. When in a complex with Bam, Otu deubiquitinates CycA, stabilizing CycA levels and thus promoting GSC daughter cell differentiation.

Here, we have generated 22 new *bam* mutants across the different binding and functional regions of *bam*. We analyzed the genetic interaction between these mutants and the *w*Mel variant of *Wolbachia* to better understand the extent of the *bam-w*Mel interaction. We are partly motivated by the observation of rapidly diversifying amino acids at *bam* across the *Drosophila* phylogeny [31–33]. *Bam* appears novel to the *Drosophila* genus [30], and we have previously hypothesized that *Wolbachia* could contribute to the rapid evolution observed at *bam* due to the observed *bam-Wolbachia* genetic interaction and the fact that both *Wolbachia* infection and positive selection at *bam* occur intermittently throughout the *Drosophila* phylogeny [11,32–34]. While there is no congruence between the patterns of infection and positive selection at *bam* across species (p = 1.0, S1 Table), we cannot disregard the potential for historic infections that may have driven positive selection at *bam*.

In this study, we find that *w*Mel genetically interacts with alanine point mutants throughout the *bam* gene, suggesting that *w*Mel rescue of *bam* mutants is not specific to any single binding/functional region of *bam*. We find no evidence for two hypotheses of how *w*Mel could rescue *bam* in a non-specific manner: There is no effect of *w*Mel on a hypomorph-like *bam* knockdown mutant, nor is there an effect of *w*Mel on the frequency of GSC mitosis that occurs upstream of *bam*. Thus, further consideration on how *Wolbachia* genetically interacts with *bam* is valuable for future endeavors that examine potential co-evolution of *bam* and *Wolbachia*.

## Materials and methods

### Design of putative *bam* hypomorph alleles

A Venus-tagged *bam* allele was designed to facilitate easy analysis of Bam protein expression and to serve as a control to the subsequent 22 Venus-tagged *bam* mutant alleles (Fig 1a), all of which are described further below. The known Bam binding residues in the CAF40 binding motif were targeted for replacement [23], producing independent alleles with mutations of L17A, M24A, and L28A. The remaining mutant alleles were designed through an alanine scanning approach [35] that targeted the known functional and/or binding regions of *bam*. Strings of charged residues were changed to alanine by choosing an alanine codon with similar usage bias in *D. melanogaster* as that of the original residue [36]. Additionally, we designed an allele

to recreate the *bam^{bw}* L255F mutation as a transgene with Venus tag. All new alleles are listed in S1 File, with the original and mutated codons.

## Generation of transgenic flies

Transgenic fly lines carrying distinct putative hypomorphic *bam*::*Venus* alleles and a control *bam*::*Venus* transgenic fly line were generated using PhiC-mediated integration [37]. Approximately 1.5 kb upstream and 700 bp downstream regions of *bam* were cloned from genomic DNA of a *D. melanogaster bam*::*Venus* line [38]. This construct is similar to that of the *D. melanogaster bam* YFP line from [11] but contains a 36bp glycine-glycine-serine linker sequence between the 3' end of the *bam* coding sequence and the Venus tag to allow for proper folding of both proteins. The *bam*::*Venus* sequence was cloned into a pMiniT 2.0 vector, using the NEB PCR cloning kit. Mutant alleles were subsequently made in the established *bam*::*Venus* pMiniT 2.0 plasmid using the NEB Q5 Site-Directed Mutagenesis Kit, with primers designed online using NEBase Changer (S2 Table) [39]. Briefly, mutagenic primers were used to amplify the *bam*::*Venus* pMiniT 2.0 plasmid using a PCR cycle of 98°C for 30 seconds; 25 cycles of 98°C for 10 seconds, 57–72°C annealing for 30 seconds (specified per primer set in S2 Table), 72°C for 3.5 minutes; and 72°C for 2 minutes. PCR products were treated with kinase, ligase, and DpnI before being transformed into competent cells. The resultant plasmids were subcloned into the pCasper\attB vector, a gift from Dan Barbash that was used by [11] for PhiC transgenesis. Plasmids were verified by sequencing.

Plasmids were injected into embryos of a *w, nos*-int; P{CaryP}attP40 *D. melanogaster* line by GenetiVision. G$_0$ flies were mated with a *w^{1118}* isogenic line and F$_1$ progeny were screened for successful transgenesis by selecting for those with mini-white eye color. Mutant lines were identified by Sanger sequencing.

All transgenic flies were assessed for function in a *bam* null background. Thus, the genotype that was assessed in all experiments was *w*; [*w^{+mc}*, *bam*::*Venus*]/+; *bam^{Δ59}*/*bam^{Δ86}*, where *bam*::*Venus* represents the control *bam* allele or one of the 22 mutant alleles. For the remainder of this paper, "transgenic females" and "transgenic *bam* mutants" refer to flies of this genotype. *Wolbachia* status of the assessed flies (uninfected or *w*Mel-infected) was established by maternal inheritance when crossing in the *bam* null alleles.

## Fertility assays of female transgenic flies

Virgin transgenic females and virgin CantonS males were collected over 48 hours and aged two days. At least 30 individual crosses of one female and two males were set up on yeast-glucose food (minimum of 30, maximum of 88 crosses were set per genotype tested). All crosses were kept in an incubator at 25°C with a 12-hour light/dark cycle. Parents were removed on the seventh day. Progeny were counted every other day over seven days, starting on the day of first eclosion. If the female parent died by the time of removal, data for that cross was discarded. Five independent fertility assays were conducted to cover all 22 mutants, with a transgenic control (*w*; [*w^{+mc}*, *bam*::*Venus*]/+; *bam^{Δ59}*/*bam^{Δ86}*) present in each assay.

The total progeny after seven days was summed and estimation statistics [40] were performed to determine first, if a fertility defect was present for a mutant, and second, if there was a *w*Mel rescue effect. Estimation statistics and plots were generated using the dabestr package in R (v4.0.1). While this offers an alternative statistical test that allows us to easily visualize effect size, we also report p-values for ease and convention. P-values were calculated through the associated website as results of a two-sided permutation t-test [41], as they were not provided in the R package.

## Nurse-cell-positive egg chamber assessment of female transgenic flies

For the ten transgenic mutants that displayed a fertility defect, an assay for nurse-cell-positive egg chambers was also performed, as has been previously done for a supplemental assessment of *bam*'s function [11,12]. Briefly, ovaries of 1–2 day old flies were fixed in 4% paraformaldehyde, rinsed and washed with PBST as described above, and then mounted in ProLong Glass with NucBlue (ThermoFisher). Ovaries were imaged on a Zeiss i880 confocal microscope at 40X and 63X (Plan-Apochromat 1.4 NA, oil), using a 405nm laser line (Cornell BRC Imaging Core Facility). Egg chambers with any identifiable large, polyploid nuclei of nurse cells indicated a successful event of Bam's differentiation function and were counted as nurse-cell-positive egg chambers. Estimation statistics were used to assess effect size and a t-test was used for statistical significance as described above for fertility of the female transgenic flies. Correlation between nurse-cell-positive egg chamber data and fertility data was performed with Kendall's rank correlation in R (v4.0.1).

## Cytological observations of the ovaries of female transgenic flies

For cytology of the ten transgenic females with fertility defects and the control females, flies were aged 2–3 days and then dissected and prepared for confocal microscopy as in [11,33]. Ovaries were dissected in ice-cold 1x PBS and fixed in 4% paraformaldehyde (Electron Microscopy Sciences) for 15 minutes. Tissue was rinsed 3x and washed 4x, respectively, with PBST (1X PBS, 0.1% Triton-X 100), blocked in PBTA (1X PBS, 0.1% Triton-X100, 3% BSA) (Alfa Aesar), and then incubated with primary antibodies overnight at 4˚C. The rinse and wash steps were repeated, followed by an overnight incubation with a goat anti-serum at 4˚C to minimize fluorescent background binding, and then a 1.5-hour incubation with secondary antibodies. The rinse and wash steps were repeated and tissue was mounted in ProLong Diamond AntiFade (Invitrogen).

The primary antibodies that were used were anti-Vasa (anti-rat, 1:20, Developmental Studies Hybridoma Bank (DSHB)), anti-Hts-1B1 (anti-mouse, 1:40, DSHB), and anti-GFP (anti-rabbit, 1:200, Invitrogen Cat No: G10362). Secondary antibodies that were used were Alexa488 (goat anti-rat, Invitrogen Cat No: A-11006), Alexa568 (goat anti-mouse, Invitrogen Cat No: A-11031), and Alexa633 (goat anti-rabbit, Invitrogen Cat No: A-21070), all at 1:500.

A Zeiss i880 confocal microscope was used for all imaging with 405, 488, and 568nm laser lines at 63X (Plan-Apochromat 1.4 NA, oil) (Cornell BRC Imaging Core Facility). Images were edited using Fiji [42].

## RNAi knockdown of *bam*

We performed reciprocal crosses between a *w*Mel-infected *nos*-Gal4 and an uninfected UAS-*bam*[HMS00029] line (a TriP *bam* shRNA line) (Bloomington *Drosophila* Stock Center lines 4442 and 33631, respectively) to generate the *bam* knockdown lines.

*Bam* expression was evaluated by RT-qPCR. RNA of four samples consisting of three 1–2 day old female flies each were prepared using the NEB Monarch Total RNA Miniprep Kit. RT-qPCR was performed with 10μL reactions (5μL Luna Universal One-Step Reaction Mix (2X), 0.5μL Luna WarmStart RT Enzyme Mix (20X), 0.4μL forward primer, 0.4μL reverse primer, 2μL template RNA, 1.7μL water), using three technical replicates and a standard curve. Reverse transcription was performed at 55˚C for 10 minutes, followed by initial denaturation at 95˚C for 1 minute, and then 40 cycles of 95˚C for 10 seconds and 60˚C for 30 seconds. Primers included 5'GCGCAATCGAAACGGAAACT3' and 5'GAGTAGCGGTGCTCCAGATC3' for *bam* and 5'CCGCTTCAAGGGACAGTATC3' and 5'CAATCTCCTTGCGCTTCTTG3' for

the reference gene *Ribosomal protein L32* (*Rpl32*) (*Rpl32* primers are from [43]). Relative expression to a wildtype *bam* control was done by the method of [44].

A fertility assay and a nurse-cell-positive egg chamber assay were performed as described above, but at 29˚C to increase the activity of the Gal4-UAS system. Estimation statistics [40] were used to compare the mean fertility of the uninfected and infected *bam* knockdown mutants. A Wilcoxon signed rank test was used to determine the statistical significance of the mean nurse-cell-positive egg chambers between the uninfected and infected *bam* knockdown mutants.

### Analysis of GSC mitosis frequency

Nurse-cell-positive egg chamber assays were performed as described above to confirm that the L255F *bam* mutation that we created as a transgene produced a partial *bam* mutant phenotype and wMel rescue like that of the original *bam*$^{bw}$ hypomorph (*w*; *bam*$^{bw}$/*bam*$^{\Delta86}$).

Then, we assessed GSC mitosis frequency in the transgenic control (*w*; [*w*$^{+mc}$, *bam*::*Venus*]/ +; *bam*$^{\Delta59}$/*bam*$^{\Delta86}$) and the transgenic L255F mutant (*w*; [*w*$^{+mc}$, *bam*$^{L255F}$::*Venus*]/+; *bam*$^{\Delta59}$/ *bam*$^{\Delta86}$) by comparing the number of actively dividing GSCs indicated by mitosis marker pH3 to the number of non-dividing GSCs. Flies were aged 2–3 days at 25˚C, at which time ovaries were dissected, immunostained and prepared for confocal microscopy, and imaged as described above.

Primary antibodies that were used were anti-Vasa (anti-rat, 1:20, DSHB), anti-GFP (anti-rabbit, 1:200, Invitrogen Cat No: G10362), and anti-pH3 (anti-rabbit, 1:200, Millipore Sigma Cat No: 06–570). The same secondary antibodies listed above (Alexa488, Alexa568, and Alexa633) were used at 1:500.

For each genotype, a Fisher's exact test was used to determine if there were more mitotically active GSCs in the *w*Mel-infected germariums than in the uninfected germariums.

### Identification of b*am* sequence amino acid divergences

*Bam* nucleotide sequences from the *D. melanogaster* Zambia population of the *Drosophila* Population Genomics Project [45] were extracted for use in the MK test. Sequences that had any ambiguous nucleotides (Ns) were removed, leaving 187 individual sequences for our analyses. The *D. simulans* FlyBase reference sequence (r2.02) was used as an outgroup. Sequences were aligned using PRANK v.170427 [46]. Nucleotide location of synonymous and nonsynonymous divergences between *D. melanogaster* and *D. simulans* were identified using a custom R script (v4.0.1) and mapped against the locations of the alanine mutations generated in this study.

## Results

### 22 mutants were generated, ten with *bam* female fertility defects

22 novel mutants were generated as *bam* transgenes with alanine replacements and a Venus fluorescent tag. They were distributed across the five functional domains/regions of the *bam* gene (Fig 1a). Each mutant was assayed for fertility defects and results were compared to a control transgenic line of a wildtype *bam* gene tagged with Venus. Mutants were characterized by comparing the average number of progeny produced by females with and without *w*Mel infection.

Of the 22 *bam* alanine-scan mutants generated, ten had fertility defects in uninfected females (Fig 2 and S1 Fig and S3 Table). These included all three of the mutants that targeted known binding residues in the CAF40 binding motif of *bam*, as well as mutants that targeted

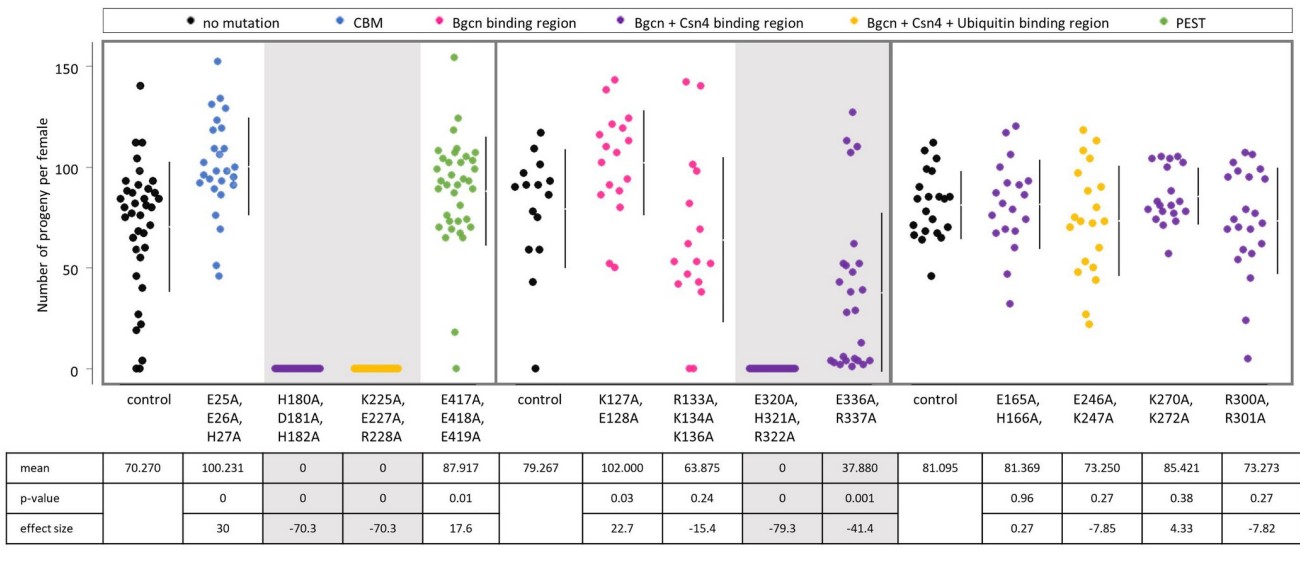

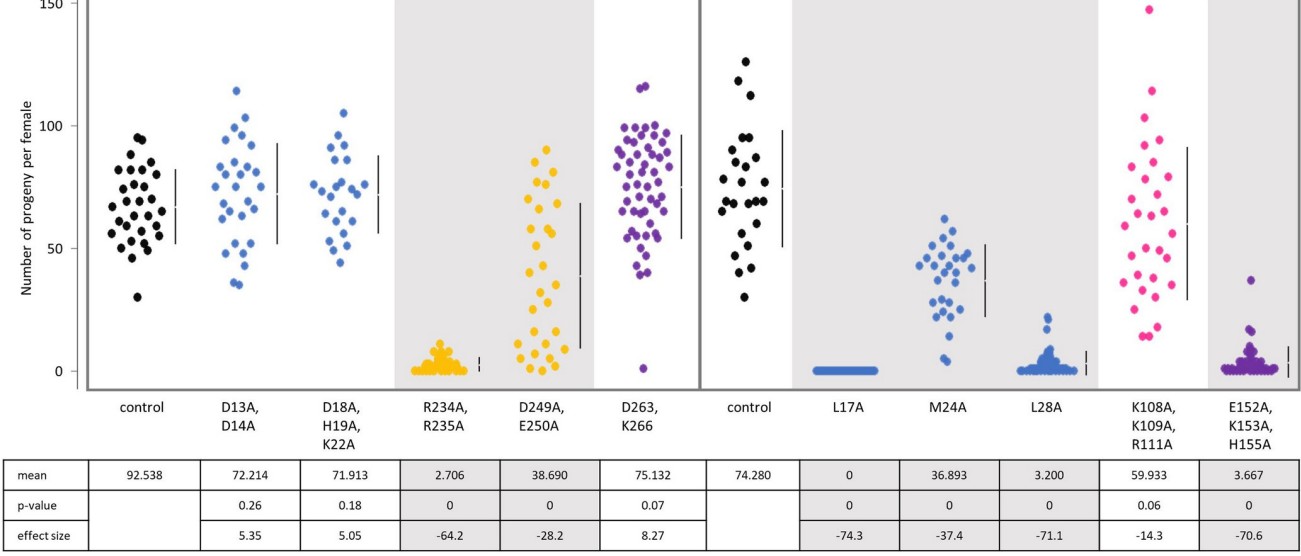

**Fig 2. Fertility of transgenic mutants compared to the respective *bam::Venus* transgenic control.** Data is colored according to the documented functional region that the mutation is located in. Below the jitter plots are the mean, two-sided permutation t-test p-value, and the average effect size between the mutant and the respective control. Gray panels indicate mutants with fertility defects (negative average effect size and p<0.05).

residues in the Bgcn, Csn4, and ubiquitin binding regions. There were four null-like fertility mutants that produced no progeny when uninfected (transgenic Venus-tagged $bam^{L17A}$, $bam^{H180A, D181A, H182A}$, $bam^{K225A, E227A, R228A}$, and $bam^{E320A, H321A, R322A}$). There were six hypomorph-like fertility mutants that produced less progeny on average than the control when uninfected. The six hypomorph-like fertility mutants included three that exhibited severe fertility defects (transgenic Venus-tagged $bam^{L28A}$, $bam^{E152A, K153A, H155A}$, and $bam^{R234A, R235A}$) and three that exhibited mild fertility defects (transgenic Venus-tagged $bam^{M24A}$, $bam^{D249A, E250A}$, and $bam^{E336A, R337A}$). The mean effect size between the number of progeny of the mutants with fertility defects compared to the control ranged from a minimum of -28.2 to a maximum of -74.3 (Fig 2 and S3 Fig).

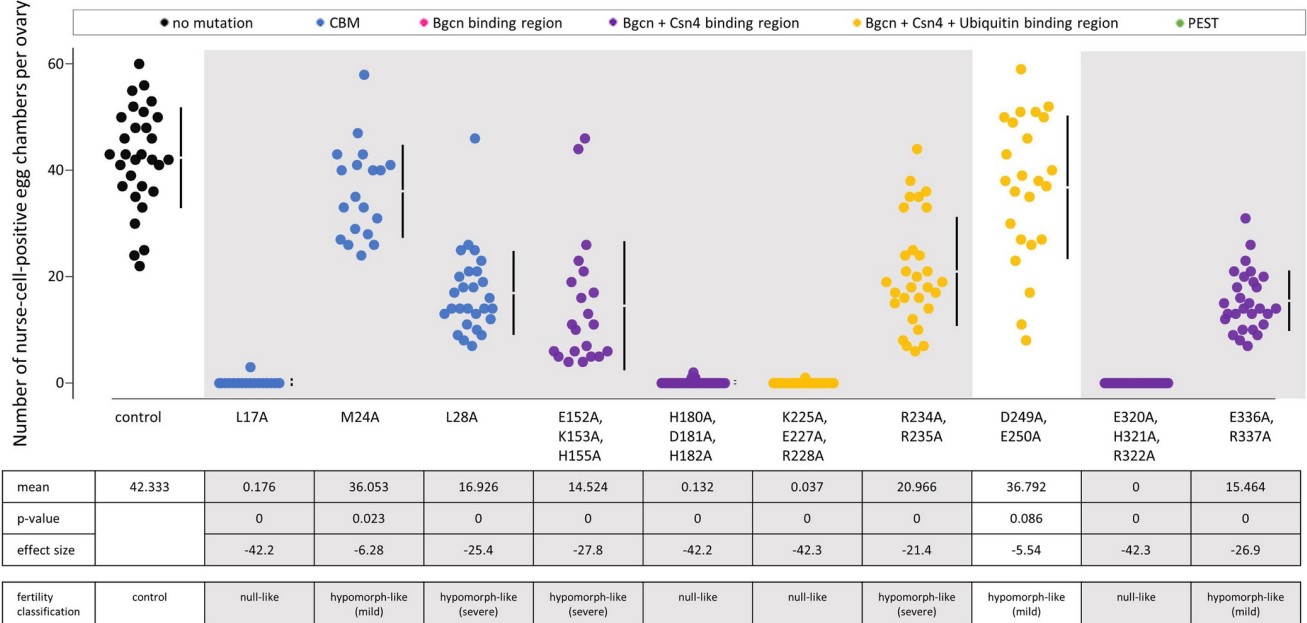

**Fig 3. Number of nurse-cell-positive egg chambers for transgenic mutants compared to the *bam*::*Venus* control.** Data is colored according to the documented functional region that the mutation is located in. Below the jitter plots are the mean, two-sided permutation t-test p-value, the average effect size between the mutant and the respective control, and the corresponding fertility classification as described in the text. Gray panels indicate mutants with nurse-cell-positive egg chamber counts that are less than the control (negative average effect size and p<0.05).

To verify that the fertility defects are a result of *bam*'s defective function in regulating GSC daughter cell differentiation, we counted the number of nurse-cell-positive egg chambers per ovary (Fig 3 and S2 Fig and S4 Table), as well as looked for the presence of tumorous germariums in uninfected flies (Fig 4). The presence of an egg chamber with large polyploid nurse cells indicates successful differentiation of a GSC daughter cell. Interestingly, for three of the four null-like fertility mutant transgenic flies (transgenic Venus-tagged *bam*[L17A], *bam*[H180A, D181A, H182A], *and bam*[K225A, E227A, R228A]) contain egg chambers with nurse cells, indicating rare occurrences of differentiation that did not translate to adult progeny in the fertility assays. Thus, these mutants are only null-like in their fertility but are not true null alleles in the assessed genetic background.

Two mutants with mild fertility defects (transgenic Venus-tagged *bam*[D249A, E250A] and *bam*[M24]) had mean nurse-cell-positive egg chamber counts that trended lower than the control, but with small effect sizes (-5.54, p = 0.086 and –6.28, p = 0.023; respectively). Overall, fertility and the number of nurse-cell-positive egg chambers correlated across the ten fertility-defective *bam* mutants (Kendall's rank correlation τ = 0.828, S3 Fig).

Our transgenic *bam*::*Venus* control had no tumorous germaria and had proper Bam expression (Fig 4a and 4b). In parallel with the novel mutants we generated, we recreated the *bam*[bw] hypomorphic mutation as a transgenic *bam*[L255F]::*Venus* mutant. As has been previously characterized for this L255F *bam* mutation in a *bam* null background, we see a tumorous germarium with some cells containing a spectrosome (Fig 4c) [8,10,24]. We also found that the overproliferating GSC-like cells in this hypomorph expressed Bam, and the lack of Bam expression in the anterior most GSCs suggested that there was no ectopic expression of Bam in the anterior part of the germarium that would cause a loss of GSCs.

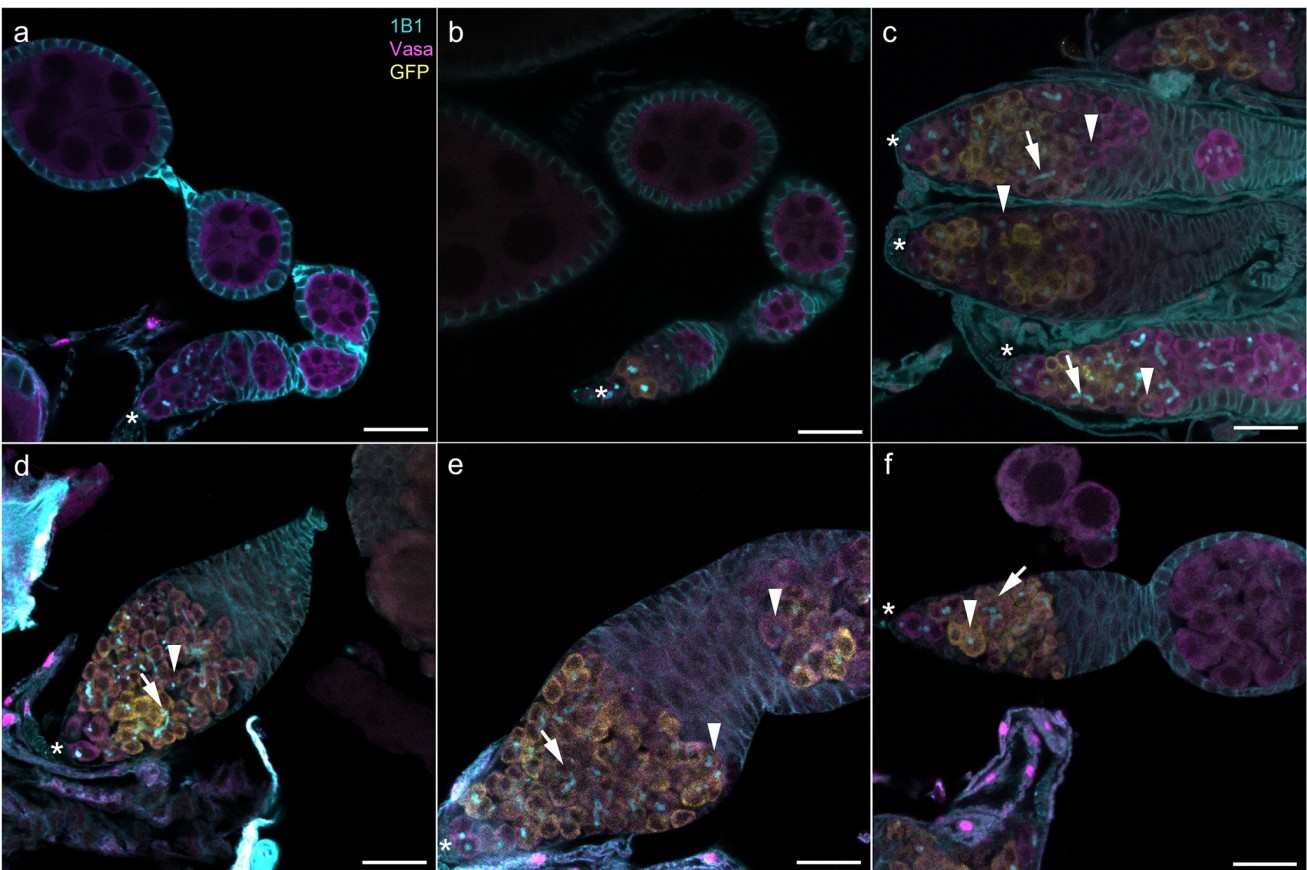

**Fig 4. Wildtype and transgenic *D. melanogaster* ovarioles stained with anti-Hts-1B1 (cyan), anti-Vasa (magenta), and anti-GFP (yellow).** (a) CantonS wildtype control. (b) Transgenic *bam::Venus* control. (c) Transgenic *bam^{L255F}::Venus* mutant. (d-f) Representative images of cytological defects observed in the transgenic *bam* mutants with fertility defects, with arrowheads indicating spectrosomes, arrows indicating branched fusomes, and asterisks indicating the apical tip of the germarium where GSCs are identifiable as cells not expressing Bam::Venus. (d) Mutant E152A, K153A, H155A. (e) Mutant K255A, E227A, R228A. (f) Mutant E336A, R337A. Scale bar for all images is 20μM.

The ten new *bam* fertility defective mutants largely displayed cytological tumorous phenotypes like that observed in the *bam^{L255F}::Venus* mutant (Fig 4d–4f and S4 Fig). Tumorous phenotypes included severely tumorous germariums with no egg chambers in the ovariole (Fig 4d); tumorous germariums with Bam expression in tumorous cells immediately outside the germarium (Fig 4e); and tumorous germariums with some tumorous egg chambers with no Bam expression (Fig 4f). Intriguingly, the *bam^{M24A}::Venus* mutant, which had a mild fertility defect, had no tumorous germariums (S4e Fig). All mutants, including the severe mutants, stained successfully with anti-GFP, indicating the presence of Bam::Venus protein.

That these ten new *bam* phenotypic mutants are distributed throughout the gene demonstrates that multiple regions of the gene and protein are critical for *bam* function. Additionally, the variation in severity of defects for mutations within one binding or functional region shows that these phenotypes are allele specific.

The twelve *bam* mutants that did not have fertility defects when uninfected were also distributed across the *bam* gene (Fig 2 and S1 Fig and S3 Table). These were the transgenic Venus-tagged *bam^{D13A, D14A}*, *bam^{D18A, D19A, K22A}*, *bam^{E25A, E26A, H27A}*, *bam^{K108A, K109A, R111A}*, *bam^{K127A, E128A}*, *bam^{R133A, K134A, K136A}*, *bam^{E165A, H166A}*, *bam^{E246A, K247A}*, *bam^{D263A, K266A}*,

$bam^{K270A, K272A}$, $bam^{R300A, R301A}$, and $bam^{E417A, E418A, E419A}$. Three of these mutants ($bam^{E25A, E26A, H27A}$, $bam^{E417A, E418A, E419A}$, and $bam^{K127A, E128A}$) had greater fertility on average than their respective control (p<0.05). Mutants without fertility defects include those with replacements at highly conserved amino acid sites and replacements at poorly conserved sites and are found throughout the gene (S5 Fig and S1 File). These results identify residues that are not critical to *bam*'s function in differentiation of GSC daughter cells in *D. melanogaster* oogenesis, at least in laboratory conditions.

## *w*Mel clearly rescues 8 of 10 fertility defective mutants

To determine if *w*Mel rescues the new *bam* mutants with fertility defects, we performed both the fertility assessment and nurse-cell-positive egg chamber assessment on the ten transgenic mutants as uninfected and infected with *w*Mel. When infected with *w*Mel, the transgenic *bam*::Venus control showed a slight increase in fertility across each independent fertility assay (p<0.05, Fig 5 and S3 Table) and in the number of nurse-cell-positive egg chambers (p<0.05,

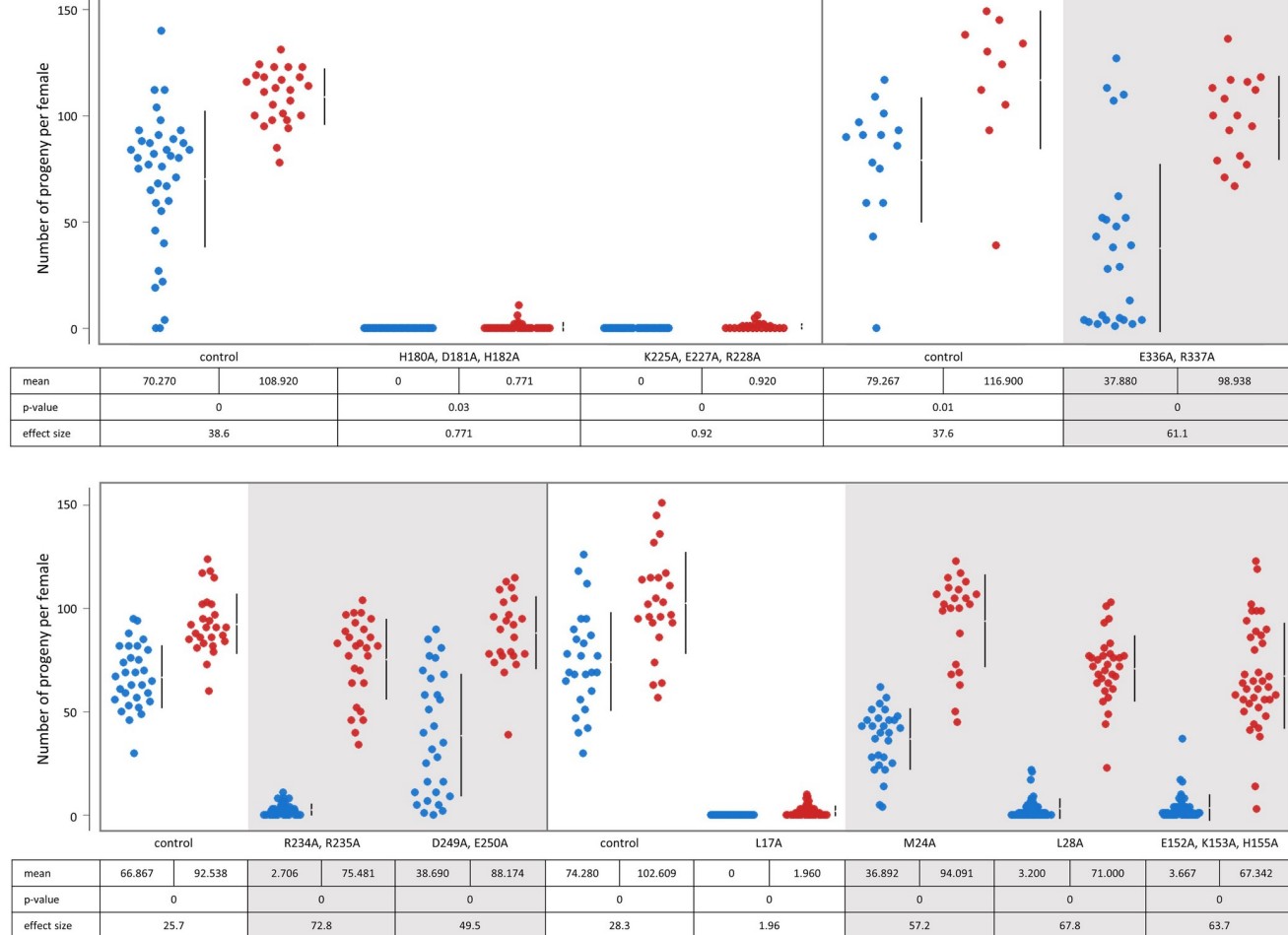

**Fig 5. Fertility of transgenic control and transgenic *bam* mutants with fertility defects as uninfected and infected with *w*Mel.** Two-sided permutation t-test p-values and the average effect size between the uninfected (blue) and infected with *w*Mel (red) data are shown below each data set. Mutant E320A, H321A, R322A was completely sterile with and without *w*Mel and thus is not shown. Gray panels highlight the mutants that are classified as having *w*Mel rescue at the fertility level due having an average effect size greater than that observed in the respective control and p<0.05.

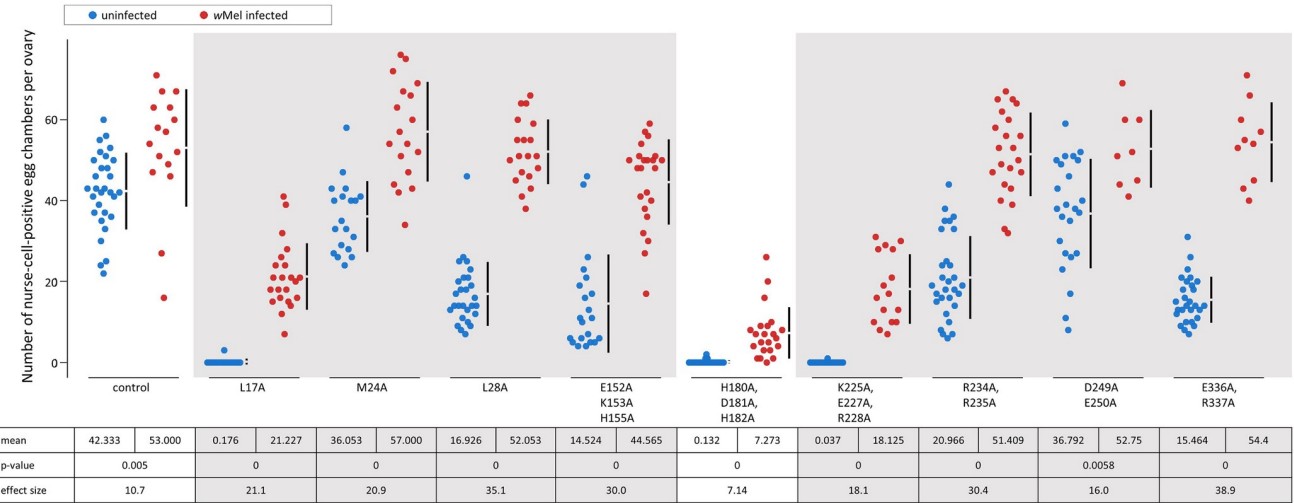

| | control | | L17A | | M24A | | L28A | | E152A K153A H155A | | H180A D181A H182A | | K225A E227A R228A | | R234A R235A | | D249A E250A | | E336A R337A | |
|---|---|---|---|---|---|---|---|---|---|---|---|---|---|---|---|---|---|---|---|---|
| mean | 42.333 | 53.000 | 0.176 | 21.227 | 36.053 | 57.000 | 16.926 | 52.053 | 14.524 | 44.565 | 0.132 | 7.273 | 0.037 | 18.125 | 20.966 | 51.409 | 36.792 | 52.75 | 15.464 | 54.4 |
| p-value | 0.005 | | 0 | | 0 | | 0 | | 0 | | 0 | | 0 | | 0 | | 0.0058 | | 0 | |
| effect size | 10.7 | | 21.1 | | 20.9 | | 35.1 | | 30.0 | | 7.14 | | 18.1 | | 30.4 | | 16.0 | | 38.9 | |

**Fig 6. Number of nurse-cell-positive egg chambers of the transgenic *bam* mutants with fertility defects, as uninfected and infected with *w*Mel.** Two-sided permutation t-test p-values and the average effect size between the uninfected (blue) and infected with *w*Mel (red) data are shown below each jitter plot data set. Mutant E320A, H321A, R322A had no nurse-cell-positive egg chambers with and without *w*Mel and thus is not shown. Gray panels highlight the mutants that are classified as having *w*Mel rescue at the nurse-cell level, due to having an average effect size greater than that observed in the control and p<0.05.

Fig 6). Thus, for a mutant to be classified with *w*Mel rescue, the increase in mean effect size had to be greater than that observed in the control.

Of the four null-like fertility mutants that produced no progeny when uninfected, three produced some progeny when infected with *w*Mel and showed an increase in nurse-cell-positive egg chambers (transgenic Venus-tagged *bam*[L17A], *bam*[H180A, D181A, H182A], and *bam*[E320A, H321A, R322A]) (p<0.05). While the slight production of progeny cannot be classified as *w*Mel fertility rescue because of the small effect size for the three mutants, there is rescue at the cytological nurse-cell level for transgenic *bam*[L17A]::Venus and *bam*[E320A, H321A, R322A]::Venus. Transgenic *bam*[H180A, D181A, H182A]::Venus trends towards rescue at the nurse-cell level, but has a small effect. The fourth null-like fertility mutant (*bam*[E320A, H32A, R322A]::Venus) did not produce progeny or nurse-cell-positive egg chambers regardless of infection status (Fig 4 and S2 Table).

All six hypomorph-like fertility mutants that had an increase in average fertility (p<0.05, Fig 5 and S6 Fig and S3 Table) and average number of nurse-cell-positive egg chambers with *w*Mel infection (p<0.05, Fig 6 and S7 Fig and S4 Table), with a mean effect size greater than that of the control). The three mutants with severe fertility defects (transgenic Venus-tagged *bam*[L28A], *bam*[E152A, K153A, H155A], and *bam*[R234A, R235A]) saw an approximate 20-fold increase in the average number of progeny per female when infected with *w*Mel. The three mutants with mild fertility defects (transgenic Venus-tagged *bam*[M24A], *bam*[D249A, E250A], and *bam*[E336A, R337A]) saw an approximate 2.5-fold increase in the average number of progeny per female when infected with *w*Mel, bringing them to a fertility level similar to that of the infected control (p>0.05 for all). The cytological data likewise showed an increase in nurse-cell-positive egg chambers in *w*Mel-infected flies that brought the total numbers to be comparable to the infected control (p>0.05 for all, except *bam*[E152A, K153A, H155A]::Venus where p = 0.042).

Overall, we find that *w*Mel rescues the reduced fertility and/or nurse-cell-positive egg chambers of eight mutants across all the documented binding regions of *bam* and throughout the predicted 3D protein structure (Fig 7) [47,48]. Of the last two mutants, the mutant that

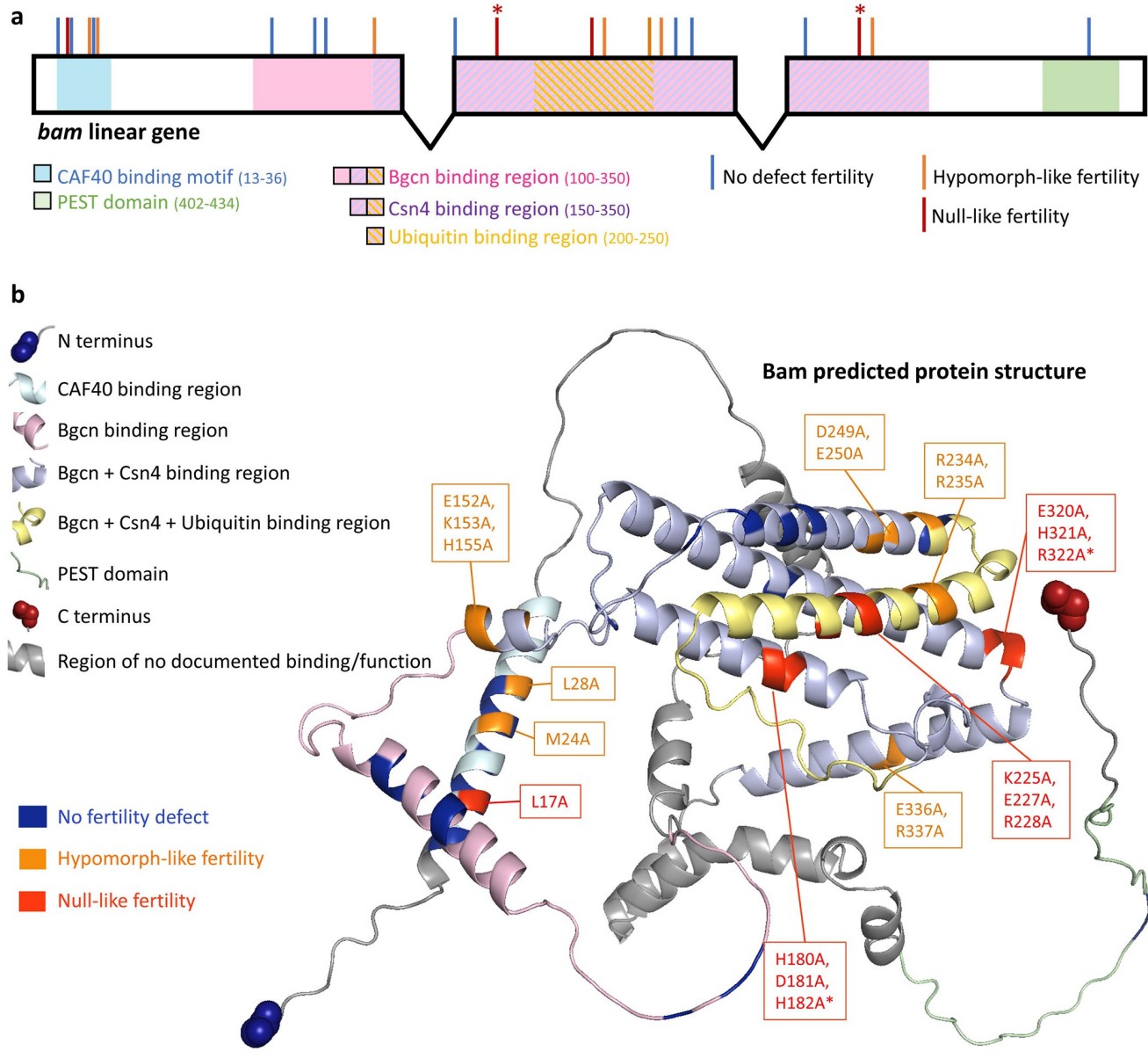

**Fig 7. The locations of the mutations generated and their effect on fertility and *w*Mel rescue across the documented binding and functional regions of *bam*.** Shown in (a) a linear schematic of the gene and (b) the predicted protein structure. * indicates fertility defect mutants in which there was no *w*Mel rescue at the fertility and the nurse-cell-positive egg chamber level.

contains some nurse-cell-positive egg chambers when uninfected trends towards rescue and the mutant that is null-like at the fertility and nurse-cell level does not. These results show that the *w*Mel rescue is not limited to the L255F *bam* mutation and that *w*Mel interacts with *bam* in a manner that is not specific to the known binding regions of the gene.

We plotted the mean progeny for all transgenic females (both uninfected and *w*Mel-infected) to easily compare the above fertility results across all mutants (Fig 8). When comparing the mean fertility for each mutant between the uninfected and *w*Mel-infected lines, we find that the data falls roughly along a logarithmic curve, suggesting there is a relationship between the severity of uninfected mutants and the increase in fertility seen with *w*Mel infection.

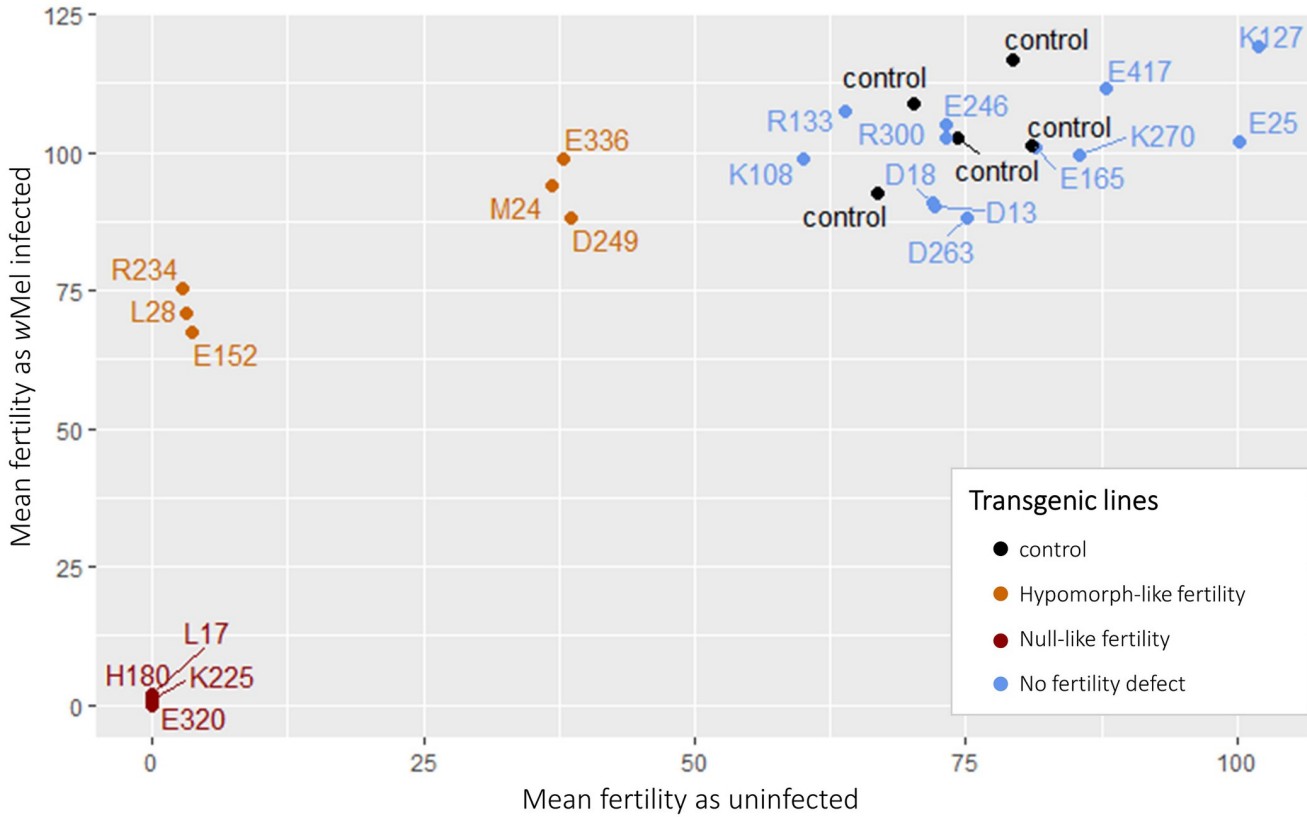

**Fig 8. Mean fertility comparison of all transgenic lines generated with and without *w*Mel.** The control lines are transgenic *bam*::*Venus* (black) and the transgenic mutants are labeled by the first residue that was changed to alanine (orange for mutants with hypomorph-like fertility defects, red for mutants with null-like fertility defects, and blue for mutants without fertility defects). Figure shows data across five independent fertility assays with five different controls.

However, we cannot discount the possibility of the increase in fertility being due to an allele-specific interaction with *w*Mel or variation in *Wolbachia* abundance.

## GSC mitosis in a *bam* hypomorphic mutant is not affected by *w*Mel

To determine if the increase in fertility and the number nurse-cell-positive egg chambers of *bam* mutants by *w*Mel is due to an increase in asymmetric division by GSCs, we quantified GSC mitosis in the *bam*::*Venus* control and the *bam*$^{L255F}$::*Venus* mutant. The *bam*$^{L255F}$ transgenic hypomorph recapitulates the severity of the endogenous *bam*$^{bw}$ hypomorph, though shows slightly reduced rescue compared to the *bam*$^{bw}$ hypomorph (S8 Fig). This difference is likely due to the differences in the genetic backgrounds of these two otherwise identical mutations.

Cells in the apical tip of the germarium that expressed the pH3 mitosis marker but not the Venus-tagged Bam protein were identified as GSCs actively undergoing mitosis (S9 Fig). The rate of GSC mitosis was not different between *w*Mel-infected and uninfected flies within both the transgenic control and the transgenic hypomorph (p = 1.0) (Table 1). Overall, the *bam*$^{L255F}$::*Venus* transgenic hypomorph had fewer GSC mitotic events compared to the transgenic control, but it was not statistically significant (p = 0.259 with *w*Mel and p = 0.448 without *w*Mel).

**Table 1. Number of mitotically active (pH3 positive) and quiescent GSCs.** GSC counts are shown for the transgenic control (*bam::Venus*) and a transgenic hypomorph (*bam^{L255F}::Venus*) when uninfected and infected with *w*Mel.

| | Transgenic control | | Transgenic *bam* hypomorph | |
|---|---|---|---|---|
| | **uninfected** | ***w*Mel-infected** | **uninfected** | ***w*Mel-infected** |
| # of germariums with pH3 positive GSCs | 5 | 9 | 2 | 4 |
| # of germariums without pH3 positive GSCs | 111 | 173 | 105 | 173 |

## *w*Mel does not rescue a *bam* knockdown phenotype

To determine if *w*Mel can rescue the phenotype of a reduced expression, but functional Bam protein in females, we tested for the rescue of a *bam* RNAi knockdown mutant by performing a reciprocal cross using the UAS-Gal4 system. We assessed both the number of egg chambers that have a nurse cell present (nurse-cell-positive egg chambers) per ovary in wildtype and *bam* knockdown mutants, as in [11,12], as well as the fertility of the knockdown mutants.

We find that the *bam* knockdown mutants have fewer nurse-cell-positive egg chambers per ovary than the wildtype lines regardless of *w*Mel status (p<0.05) (Fig 9a and S10 Fig). *Bam* knockdown mutants with *w*Mel had fewer nurse-cell-positive egg chambers on average than *bam* knockdown mutants without *w*Mel (p<0.05) but had similar levels of fertility over a period of seven days (p = 0.69, Fig 9b). This difference is likely due to the narrow window of

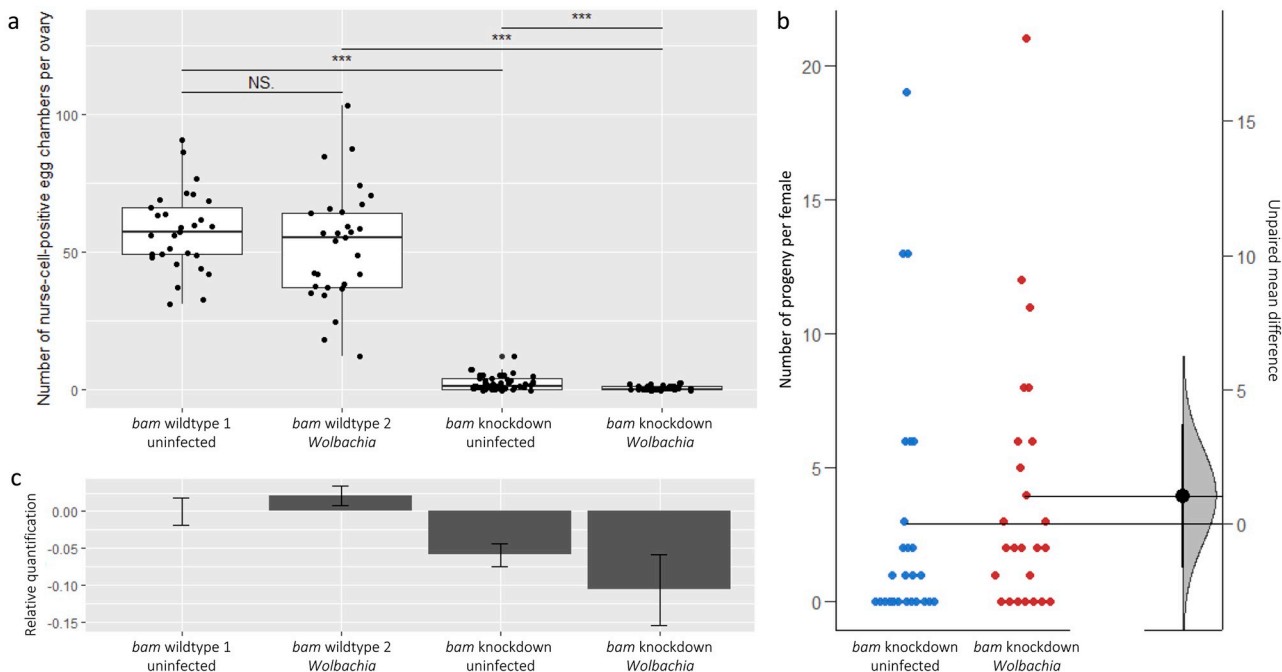

**Fig 9. *Bam* knockdown by RNAi in the UAS-Gal4 system.** *Bam* wildtype 1 and 2 are the UAS and Gal4 stocks, respectively; *bam* knockdowns are the nos-Gal4/+; UAS-*bam*^{HMS00029}/+ reciprocal F1 progeny of the stocks. (a) The number of nurse-cell-positive egg chambers per ovary. *** = significant at p<0.001; NS = not significant. (b) Fertility of the *bam* knockdown mutants as uninfected and *w*Mel-infected. Jitter plots (left) show the raw data as dots with horizontal lines representing the mean values of each dataset. The mean difference between uninfected and infected lines are plotted as a bootstrap sampling distribution with a 95% confidence interval (right). (c) *Bam* expression on a log2 scale relative to *bam* wildtype 1. Each bar represents the dataset of four samples consisting of three female flies, and shows the mean value with standard error.

time captured in the nurse-cell-positive assay compared to the fertility assay. These results show that *w*Mel does not rescue the tumorous mutant phenotype or the fertility of a *bam* knockdown mutant.

Additionally, there is no increase in *bam* expression in the knockdown mutant with *w*Mel compared to the uninfected knockdown mutant (Fig 9c). This shows that *w*Mel does not increase the expression levels of *bam*, consistent with the lack of rescue that we observe. All together, these results suggest that *w*Mel rescue is specific to partial-loss-of-function protein coding *bam* mutants.

## Discussion

The critical reproductive gene *bam* is one of a few genes that has been shown to genetically interact with *Wolbachia* in *D. melanogaster* [11,18]. In female and male adults, *bam*'s molecular role is to bring together other proteins into complexes, which then carry out functions vital to the maintenance or progression of gametogenesis. *Bam*'s central and extensive role in early reproduction, as well as *Wolbachia*'s potential for evolutionary impact motivates a question about the specificity of the *bam-Wolbachia* interaction. Here, we first discuss our results considering the molecular nature of the *bam-Wolbachia* interaction and then follow up with a discussion on implications for the *D. melanogaster* host and *w*Mel *Wolbachia* variant from an evolutionary perspective.

We targeted *bam*'s role of protein binding and minimized affecting the overall conformation of the Bam protein by replacing charged residues with alanine throughout *bam*'s documented binding regions [35]. We find no specificity to the presence/absence of rescue or the magnitude of fertility rescue for the different functional regions of the linear *bam* sequence or with the AlphaFold-predicted Bam protein structure. The strongest evidence for this comes from our finding that *w*Mel rescues fertility defects of the known CAF40 binding residues (*bam*[M24A] and *bam*[L28A]), as well as fertility defects of several mutations not involved in CAF40 binding. This suggests that *w*Mel does not rescue the specific interaction between Bam and a single one of its known protein partners in *bam* fertility mutants. Of note, however, is the possibility that the fertility defects have arisen from a disruption of Bam protein structure despite the practice to mitigate this with an alanine-scanning approach. Additionally, it is possible that the rescued mutants' amino acid sites are involved in a direct interaction between Bam and an unmapped binding partner (e.g., Otu [25]).

The severe and mild mutants were largely rescued to the level of the infected control females. Thus, the severity of the mutation likely plays a large role in *w*Mel's ability to have a substantial impact on host reproductive success. The biochemical nature of the allele interactions and the effect of *w*Mel titer on host reproductive success are areas to be further investigated. Additionally, some egg chambers were noticeably mutant in their cytology, which may explain discrepancies between the nurse-cell-positive egg chamber data and the fertility data. As *bam* also plays a role in mitotic divisions of the germline [9], nurse-cell-positive egg chambers may be present, but with an incorrect number of nurse cells.

We tested two alternative hypotheses for how *w*Mel could be rescuing *bam* fertility mutants. The first proposes that *w*Mel induces a fertility rescue by affecting *bam*'s expression. It has previously been shown that there is no difference in *bam* expression in the *bam*[bw] hypomorph as uninfected and infected with *w*Mel [11]. Here, we tested the ability of *w*Mel to rescue the mutant phenotype of a wildtype *bam* gene that is reduced in expression via RNAi. Importantly, we have a knockdown mutant that displays the hypomorph phenotype of tumorous germariums with some wildtype egg chambers. If the knockdown phenotype is too severe, *w*Mel may have no effect as evidenced by the null-like fertility mutant generated in this study (transgenic

*bam*$^{E320A, H321A, R322A}$::*Venus*) and *w*Mel's lack of interaction with the *bam* null mutant documented previously [11]. With the hypomorph-like-phenotype *bam* knockdown mutant (*nos*-Gal4/+; UAS-*bam*$^{HMS00029}$/+), we find that *w*Mel does not rescue the number of nurse-cell-positive egg chambers or affect the fertility of female flies; it also does not increase the expression level of *bam*.

The second hypothesis proposes that *w*Mel increases the rate of GSC mitosis, which increases the output of reproduction. Previous studies have shown *w*Mel to interact with GSCs, including preliminary research in a *mei-P26 D. melanogaster* mutant that shows increased GSC mitosis in *w*Mel-infected germariums compared to uninfected germariums [49]. Other evidence of *w*Mel's interaction with GSCs includes research on the *Sxl*$^{f4}$ *D. melanogaster* mutant that has shown that *w*Mel restores the number of GSCs that were lost due to ectopic *bam* expression [22] and the necessity for *w*Mel for GSC maintenance in filarial nematodes [50].

We find that *w*Mel does not increase the rate of GSC mitosis in the *bam*$^{L255F}$::*Venus* hypomorph, and we do not observe any obvious ectopic Bam expression in the GSC niche in any of our transgenic *bam* mutants. These results are consistent with what we see with the *bam* knockdown results, as an increase in GSC mitosis rate would likely be reflected in an increase in progeny in the *w*Mel-infected *bam* knockdown females.

Thus, while it is possible for there to be a subtle *w*Mel-GSC interaction that we have not yet assessed, collectively, our results suggest that a large component of the *w*Mel *bam* interaction is specific to mutant phenotypes that are the result of a disruption of *bam* function. Biochemical assays are needed to firmly conclude that the *w*Mel *bam* interaction is specific to *bam* functions, but it is notable that this proposal is akin to that suggested for *Wolbachia's* interaction with *Sxl* [51]. The observations that *Wolbachia* cannot rescue the reduced egg count of a *snf* mutant, which has no *Sxl* expression in the germline, and a *Sxl*$^{f5,fHa}$ mutant, which has a transposon insertion in the second intron [21] parallels the *bam* knockdown results of our study.

It is possible that the fertility rescue of *bam* mutants occurs through the same (unknown) mechanism by which the rescue of *Sxl* mutants occurs. This would be a process beyond the documented restoration of GSC number in *Sxl*$^{f4}$ mutants by *Wolbachia* factor TomO, as it was found that TomO is not sufficient to restore fertility of the *Sxl*$^{f4}$ mutants [22].

Bam and Sxl have been proposed to partner together to repress the translation of the stem cell maintenance factor *nos* to determine GSC daughter cell fate [27,28] and to maintain female sexual identity in the adult germline [52]. Alternatively, it is possible that *Wolbachia* interacts with *bam* and *Sxl* independently. Bam has a number of other genetic and physical interacting partners that offer functional pathways for which *Wolbachia* could be interacting with. For example, Bam promotes differentiation of GSC daughter cells by facilitating deubiquitination of CycA with Otu [25].

Regardless of the manner in which *w*Mel molecularly interacts with *bam*, that *w*Mel increases fertility of almost all *bam* protein-coding mutants generated in this study has broader implications for *bam* and *Wolbachia's* population and evolutionary dynamics. A recent study modeling potential *bam-Wolbachia* interactions suggests that *w*Mel's ability to mediate the negative fitness consequences of protein-coding mutants like that which is observed here, is not enough to generate the magnitude of positive selection currently observed at *bam* [53]. The nonsynonymous amino acid replacements that are prevalent in *bam* across several *Drosophila* species [31–33] may instead be due to arms race dynamics between *bam* and *Wolbachia*, an evolutionary process independent of *Wolbachia*, or some combination of the two.

For *Wolbachia*, the increased reproductive success for almost all *bam* protein-coding mutants with *w*Mel infection is an addition to *w*Mel's arsenal of promoting its transmission and spread through populations. It is particularly advantageous for *w*Mel, as, unlike some

other *Wolbachia* variants, it exhibits minimal cytoplasmic incompatibility in field-caught *D. melanogaster* [19]. Ultimately, that *w*Mel rescues fertility defects of multiple *bam* protein-coding mutants, as well as evidence that *Wolbachia* rescues reproductive defects of other germline genes such as *Sxl* and *mei-P26*, may allow for opportunities for an obligate symbiotic relationship between *D. melanogaster* host and *w*Mel to develop.

## Supporting information

**S1 Fig. Full estimation plots for fertility data of transgenic mutants compared to the *bam*::*Venus* transgenic control.** Fertility assays were done in five independent rounds. Below the jitter plots are the resampled bootstrap sampling distributions, with the mean differences represented by the black dots and the 95% confidence intervals represented by the vertical lines. Mean and sample size for each sample are listed in S3 Table.
(TIF)

**S2 Fig. Full estimation plots for nurse-cell-positive egg chamber data of transgenic mutants compared to the *bam*::*Venus* transgenic control.** Below the jitter plots are the resampled bootstrap sampling distributions, with the mean differences represented by the black dots and the 95% confidence intervals represented by the vertical lines. Mean and sample size for each sample are listed in S4 Table.
(TIF)

**S3 Fig. Correlation of mean fertility and mean number of nurse-cell-positive egg chambers per ovary.** Each point represents the mean data for a fertility defective transgenic *bam* mutant line, with uninfected data in blue and infected data in red. Kendall's rank correlation $\tau = 0.828$.
(TIF)

**S4 Fig. Germariums of controls and all transgenic *bam* mutants with fertility defects when uninfected with *w*Mel.** Stained with anti-Hts-1B1 (cyan), anti-Vasa (magenta), and anti-GFP (yellow). (a) CantonS, (b) *bam*::*Venus* control, (c) *bam*$^{L255F}$::*Venus*, (d) *bam*$^{L17A}$::*Venus*, (e) *bam*$^{M24A}$::*Venus*, (f) *bam*$^{L28A}$::*Venus*, (g) *bam*$^{E152A, K153A, H155A}$::*Venus*, (h) *bam*$^{H180A, D181A, H182A}$::*Venus*, (i) *bam*$^{R234A, R235A}$::*Venus*, (j) *bam*$^{D249A, E250A}$::*Venus*, (k) *bam*$^{K255A, E227A, R228A}$::*Venus*, (l) *bam*$^{E320A, H321A, R322A}$::*Venus*, (m) *bam*$^{E336A, R337A}$::*Venus*. Scale bar is 20μM for all images.
(TIF)

**S5 Fig. Location of transgenic *bam* alanine mutants with respect to divergent amino acids between *D. melanogaster* and *D. simulans*.** Location of alanine mutations are represented by vertical lines, with fertility defect mutants in black and non-defect mutants in grey. Black dots represent the location of divergent amino acids between *D. melanogaster* and *D. simulans*, with synonymous divergences in the top row and nonsynonymous divergences in the bottom row. Background colors correspond to known functional and/or binding regions in *D. melanogaster*.
(TIF)

**S6 Fig. Full estimation plots for fertility data of transgenic fertility-defective females, compared between uninfected and *w*Mel-infected.** Below the jitter plots are the resampled bootstrap sampling distributions, with the mean differences between the uninfected and infected data represented by the black dots and the 95% confidence intervals represented by the vertical lines. Mean and sample size for each sample are listed in S3 Table.
(TIF)

**S7 Fig. Full estimation plots for nurse-cell-positive egg chamber data of transgenic fertility-defective females, compared between uninfected and *w*Mel-infected.** Below the jitter plots are the resampled bootstrap sampling distributions, with the mean differences between the uninfected and infected data represented by the black dots and the 95% confidence intervals represented by the vertical lines. Mean and sample size for each sample are listed in S4 Table.
(TIF)

**S8 Fig. Comparison of original L255F mutant (*bam^bw^*) and the transgenic L255F mutant with a Venus tag (*bam^L255F^::Venus*) by the number of nurse-cell-positive egg chambers per ovary.** ** = significant at p<0.01; *** = significant at p<0.001; NS = not significant.
(TIF)

**S9 Fig. GSC mitosis in the transgenic *bam::Venus* control and transgenic *bam^L255F^::Venus* hypomorph.** (a) No GSC mitosis and (b) active GSC mitosis in the *bam::Venus* control germarium. (c) No GSC mitosis and (d) active GSC mitosis in the *bam^L255F^::Venus* hypomorphic mutant.
(TIF)

**S10 Fig. Ovaries of a *bam* wildtype and *bam* RNAi knockdown fly.** (a) Ovaries of a *bam* wildtype fly. All egg chambers are results of successful Bam function. (b) Mutant ovaries of a nos-Gal4/+; UAS-*bam^HMS00029^*/+ uninfected fly. Arrows indicate nurse-cell-positive egg chambers that represent successful Bam function. Scale bar for both images is 100μM.
(TIF)

**S1 Table. Presence or absence of positive selection at *bam* and known *Wolbachia* infection status for *Drosophila* species.** Selection at *bam* is informed by McDonald-Kreitman tests from [30]. *Wolbachia* infection status for each species is cited individually in the table. N/A indicates there were no publications on infection status were found. Species that have evidence for both *Wolbachia* infection and absence reflect different stocks or geographical samples studied and were not included in the subsequent statistical analysis. In total, there are 4 species of *Wolbachia* infection and positive selection; 3 species of *Wolbachia* absence and positive selection; 3 species of *Wolbachia* infection and no positive selection; 2 species of *Wolbachia* absence and no positive selection. Fisher's exact test results are p = 1.0.
(XLSX)

**S2 Table. Mutagenesis primers used to generate new *bam* mutants.** Each row lists the corresponding primer sequences and annealing temperature used in the mutagenesis PCR step of the cloning procedure to generate the respective amino acid mutant.
(XLSX)

**S3 Table. Mean adult progeny per female seven days after first progeny eclosed, as uninfected and infected with *w*Mel.** *Bam* alleles are listed in order of location of mutations from N- to C-terminal, with mean fertility of and sample size of surviving females at the end of the assay. The corresponding assay in which the transgenic flies were evaluated is indicated by the fertility assay round.
(XLSX)

**S4 Table. Mean number of nurse-cell-positive egg chambers per ovary, as uninfected and infected with *w*Mel.** *Bam* alleles are listed in order of location of mutations from N- to C-terminal, with mean number of nurse-cell-positive egg chambers per ovary and sample size.
(XLSX)

**S1 File. Summary of all generated transgenic mutants.** Included for each mutant are the amino acid site(s) and the corresponding functional and/or binding region; the original codon and residue; the number of species that the residue is conserved across from the seven species (*D. melanogaster*, *D. simulans*, *D. yakuba*, *D. santomea*, *D. teissieri*, *D. erecta*, and *D. eugracilis*) multiple alignment of [38]; the alanine codon that replaced the original codon sequence; and the results of the fertility and nurse-cell-positive egg chamber experiments.
(XLSX)

**S1 Data. Underlying data for fertility and nurse-cell-positive egg chamber figures.**
(XLSX)

## Acknowledgments

We thank Dan Barbash for sharing the pCasper\attB vector and are grateful to Eric Alani for advice on generating alanine-scanning mutants. We thank Mariana Wolfner and Andy Clark for insightful discussions and review of the manuscript. We also thank Jackie Bubnell, Catherine Kagemann, and Luke Arnce for thoughtful feedback throughout this project.

## Author Contributions

**Conceptualization:** Miwa Wenzel, Charles F. Aquadro.

**Formal analysis:** Miwa Wenzel.

**Funding acquisition:** Charles F. Aquadro.

**Investigation:** Miwa Wenzel.

**Methodology:** Charles F. Aquadro.

**Supervision:** Charles F. Aquadro.

**Writing – original draft:** Miwa Wenzel.

**Writing – review & editing:** Charles F. Aquadro.

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
