## [Decision Letter · Decision Letter 0]

1 May 2023

Dear Dr Wenzel,

Thank you very much for submitting your Research Article entitled 'Wolbachia rescue of several Drosophila melanogasterbag of marbles protein-coding mutants' to PLOS Genetics.

The manuscript was fully evaluated at the editorial level and by two independent peer reviewers, who are both experts in the field. The reviewers appreciated the attention to an important problem, but raised some substantial concerns about the current manuscript. Based on the reviews, we will not be able to accept this version of the manuscript, but we would be willing to review a much-revised version that incorporates their constructive criticism to better present the current work into a larger biological framework; one reviewer commented that they thought the work as presented was quite narrow. We cannot, of course, promise publication at that time but we will consider the revision in context of these reviews and your revisions.

If you decide to revise the manuscript for further consideration at PLOS Genetics, please aim to resubmit within the next 60 days, unless it will take extra time to address the concerns of the reviewers, in which case we would appreciate an expected resubmission date by email to plosgenetics@plos.org.

We are sorry that we cannot be more positive about your manuscript at this stage. Please do not hesitate to contact us if you have any concerns or questions.

Yours sincerely,

Harmit S. Malik

Academic Editor

PLOS Genetics

Gregory P. Copenhaver

Editor-in-Chief

PLOS Genetics

Reviewer's Responses to Questions

**Comments to the Authors:**

Reviewer #1: This paper follows previous work from the lab published in 2015 in the same journal. That paper showed Wolbachia infection also partially rescues the sterility of bam hypomorphic mutants. Here they continue this line of research by testing rescue of fertility defects across 22 new bam mutants in which the most compelling data of the paper indicate Wolbachia rescue a variety of mutations that reduce fertility of uninfected mutants in the range of 1-50% fertility reductions, but mostly not above or below. The results also support that no single function stands out for how Wolbachia rescues bam function.

Major comments:

Line 74: The connection between Sxl and bam is an intriguing hypothesis. That would be fun to test!

Line 132: The mutants were created with a Venus fluorescent tag. Mention if this is a variable in the experiment that is controlled for or not.

Line 178: How can it be surmised that a fertility defect does not result from a vigor defect of the mutant (e.g. do mutant females escape from the two mating males they are paired with so that the female cannot lay enough eggs)? For instance, if the males were removed from the assay, maybe some of the mutant females would have no fertility defect. That could significantly change the interpretations. I recognize there might be some wiggle room here about semantically defining fertility and vigor competition with males, but it would be good to clarify these options in the text and use generalist language to reflect their biological differences. The ovary data is not yet convincing yet to firmly conclude the bam mutation only works through impact on ovaries (See below)

Line 223+: Does an ovary defect technically and directly translate to a fly fertility defect? The work can provide quantitative data that would support a statistical correlation between the average fly fertility effect versus the average (or %) defect in the ovary to build the case. Currently, single to a few images are presented; good evidence would be quantitative and backed by statistics. This seems important to resolve so the above alternative explanation about vigor defects could be rendered a bit less likely.

Line 314+: The mutants without a fly fertility defect would be great to show that ovary cytology is normal and thus correlates well with the fly mutants with normal fertility. Is this available?

Minor comments:

Figure 1: Great figure and good to connect with a similar figure at the end of the article.

Line 63: For equity and since most articles do not have authors names, restructure the sentence to be agnostic to author names" to avoid unintentional bias.

Line 271+: The alternative explanation for the widespread effects of the mutations

on fertility is a disruption of protein structure. This can be added here and other spots.

Figure 5: Which sites are exposed on the surface versus internal to the protein? A view of the protein surface with color coded sites will help.

Figure 7 et al: This will probably be edited in production, but just in case, remove the grey box outlines around the figure(s).

Reviewer #2: In this piece, Wenzel and Aquadro build on some really nice past work by further evaluating interactions between the protein-coding gene bag of marbles (bam) (in melanogaster) and endosymbiotic Wolbachia (wMel) bacteria. Briefly, bam has an important role during gametogenesis—in females bam regulates and induces daughter cell differentiation, while in males it initiates terminal differentiation of spermatogonia. Null bam mutants of both sexes are sterile, and partial loss-of-function mutants have lower female fertility. Past work has demonstrated that Wolbachia infections in females can rescue fertility reductions. While I could go on about the very interesting specific biology, this paper focuses on evaluating the specificity of the bam-Wolbachia interaction using 22 novel bam mutants. Some major results include no apparent specificity of rescue for the different functional bam regions or with predicted Bam protein structure, with rescue observed for all cases of partial fertility reduction. The authors also find no evidence that Wolbachia interacts with germline stem cells in bam mutants. These are nice findings that further develop this system.

I think this is an interesting paper, and my major comments are related to better connecting this work to relevant patterns in nature and to the possible influence of Wolbachia on bam evolution, which seems very unclear to me at this stage. I also think that including data on Wolbachia abundance (versus relying on infected/uninfected alone) would improve the interpretation of patterns. I don’t think the data support this being a general thing for “Wolbachia”, but rather a wMel-specific interaction. This is worth noting since Wolbachia and bam are diverse, and since Wolbachia infect most insects, but bam seems to be specific to Drosophila (i.e., the results are not as general as pitched).

A few bigger points:

bam evolution: The authors lay out at line 117 that rapid evolution of bam across several Drosophila motivates their interest here, with the potential for Wolbachia to have played a role. At a minimum, the authors need to summarize those patterns in more depth here. If only a handful of species have been assessed, it might be worth taking a look at new data (e.g., Suvorov et al. 2022, Current Biology). There are clades that are Wolbachia uninfected, and my thought is that with more data one could pretty quickly better test the hypothesis of a plausible Wolbachia role in rapid bam evolution. The caveat here is that Wolbachia seem to regularly switch hosts, so any uninfected species may have been infected in the past. All functional bam-Wolbachia work has been in melanogaster (I think), making this sort of analysis even more useful since we don’t know if the interaction is wMel specific. For clarity, I think that adding more context is likely to be sufficient here, but adding an additional analysis could increase impact (but is not necessary).

Also, is bam itself specific to Drosophila? I ask because I took a (very) quick look at some data, and I don’t see obvious orthologs across several holometabolous insects. This along with rapid evolution in Drosophila is really interesting, and I think other evolutionary-leaning readers will appreciate more context, even if it is redundant with prior work. I did not have time to revisit all past papers to confirm what is known (or not), and most readers won't either.

Relevance of bam-Wolbachia interactions in natural populations: We know that Wolbachia must increase host fitness to initially spread in populations (Hoffman et al. 1990), and positive fecundity effects have been proposed (Fast et al. 2011, but see my comment below; Weeks et al. 2008). I think this piece could be improved by connecting the interesting genetics here to its potential role in a natural context, if there is any. As the authors know, Wolbachia are often imperfectly transmitted. This breaks up any linkage disequilibrium between host variation and Wolbachia each generation. wMel occurs at variable intermediate frequencies in melanogaster. I’m left wondering if this interaction is likely to be meaningful for Wolbachia spread in nature? Broadly, I think the piece could be improved by connecting the interesting molecular genetic results here to relevant evolutionary genetic considerations.

Hypotheses for how Wolbachia could be rescuing bam fertility mutants: While I think the Discussion is really great and thoughtful, I want to make one relevant point that the authors might consider. The hypothesis that Wolbachia (wMau) increases the rate of GSC mitosis is less supported than indicated here. Meany et al. 2019, Evolution demonstrates a lack of any wMau fecundity effect across age and D. mauritiana backgrounds. Perhaps more importantly, this paper also demonstrates the four-fold fecundity effect reported by Fast et al. (2011) is not theoretically plausible given known parameter estimates. It’s unclear to me how the other main results of Fast et al make much sense once you consider this. I don't believe anyone has been able to replicate the Fast et al. result.

Wolbachia variant and titer: What Wolbachia variant is being used here (sorry if I missed this)? wMel or wMel CS? Is it a sequenced variant, and if so what wMel clade (Bergmann et al. 2012, PG)? More importantly, do the authors have any data on Wolbachia abundance in host tissues? Almost all Wolbachia effects on hosts depend on abundance/localization, and that is probably true here. The question is how much abundance varies among infected genotypes. Considering Wolbachia abundance in a quantitative way is likely to improve the interpretation of the results here, but I fully realize these data may not exist.

Other points:

Is there any evidence that “Wolbachia” generally interacts with bam, or are all results specific to wMel and melanogaster? If so, is all work on one wMel variant, or is it known that wMel variation contributes to variation in relevant phenotypes here? I think it is worth being clear about this. Without knowing interacting genomic factors on the Wolbachia side, it seems impossible to speculate much about the generality of this interaction from sequence analysis alone, and I don’t think other Wolbachia have been introgressed to look at phenotypes?

Abstract: Consider being more clear about the specific Wolbachia being assessed (wMel presumably) and that the approach is using transgenics.

Line 57: It is worth noting that CI reduces egg hatch, but the strength of this reduction can vary widely (i.e., many progeny remain viable depending on the system/context; see Dylan Shropshire’s work).

Figure 1: The authors might consider indicating patterns of selection across bam. This could easily be integrated into 1A. Presumably the focal binding regions are *relatively* conserved?

Line 178: Do you happen to have egg lay, hatch, larval counts, or any other measures? Number of progeny is fine, but I’m personally interested in if you can tease apart any development-specific rescue.

Figure 2: Colors will be challenging for colorblind readers.

Figure 2: What is the interpretation of fertility enhancement relative to controls in multiple cases? Maybe there isn’t one.

Line 343: Wow. I think a 20-fold increase is striking and could rise to the level of being noted in the abstract.

Figure 4: It seems plausible that Wolbachia abundance varies among infected genotypes, and if the authors have those data, it would be good to know if the degree of recovery is a function of variable Wolbachia abundance in tissues or even whole bodies.

Line 377: Or variation in Wolbachia abundance.

Line 473: This result cannot be replicated as noted above.

**Have all data underlying the figures and results presented in the manuscript been provided?**

Reviewer #1: Yes

Reviewer #2: **No: **I'm selecting "no" because I cannot confirm that the raw data are available.

PLOS authors have the option to publish the peer review history of their article (what does this mean?). If published, this will include your full peer review and any attached files.

Reviewer #1: No

Reviewer #2: No

---

## [Decision Letter · Decision Letter 1]

6 Oct 2023

Dear Dr Wenzel,

We are pleased to inform you that your manuscript entitled "Wolbachia infection at least partially rescues the fertility and ovary defects of several new Drosophila melanogaster bag of marbles protein-coding mutants" has been editorially accepted for publication in PLOS Genetics. Congratulations!

Yours sincerely,

Harmit S. Malik

Academic Editor

PLOS Genetics

Gregory P. Copenhaver

Editor-in-Chief

PLOS Genetics

Comments from the reviewers (if applicable):

Reviewer's Responses to Questions

**Comments to the Authors:**

Reviewer #1: Thanks to the authors for the excellent care and follow-up work to the questions. I was impressed with the correlation between fecundity and ovary integrity for the bam mutants, the controlled aspects of the Venus tag, etc. The final area that would benefit from a tad more work is using other structural prediction programs besides alpha-fold to predict exposed versus internal sites of bam. Then they can map amino acid sites to predicted protein locations.

1. In a protein, hydrophobic amino acids are likely to be found in the interior, so recommend use of a simple hydrophobicity index to make some predictions and connections.

2. Try predictive models besides alpha-fold.

Reviewer #2: -My apologies to the authors for my slow response time that resulted from travel and university obligations. The authors have addressed my major concerns by revising the text and adding additional analyses. I think this version is much better in terms of placing the work in a broader (evolutionary) context. I have a few additional comments that may improve the paper. All are minor. My new comments have a “-“ in front of them.

-Prior points that remain relevant:

Original comment - bam evolution: The authors lay out at line 117 that rapid evolution of bam across several Drosophila motivates there interest here, with the potential for Wolbachia to have played a role. At a minimum, the authors need to summarize those patterns in more depth here. If only a handful of species have been assessed, it might be worth taking a look at new data (e.g., Suvorov et al. (2022, Current Biology). There are clades that are Wolbachia uninfected, and my thought is that with more data one could pretty quickly better test the hypothesis of a plausible Wolbachia role in rapid bam evolution. The caveat here is that Wolbachia seem to regularly switch hosts, so any uninfected species may have been infected in the past. If I am correct, all bam-Wolbachia work has been in melanogaster, making this sort of analysis even more useful since we don’t know if the interaction is wMel specific.

Response from authors: We have greatly expanded on the evolutionary perspective of this

interaction. This includes additional text in the introduction and in the discussion (L123 - L131).

We had internally performed the analysis specifically suggested by this reviewer in this

comment, but had never published the results and thank the reviewer for the encouragement to

do so here. This is now at L130. We have included the caveat mentioned about unknown past

infections. (In general, we have thought about this extensively and expect a separate publication

on our work in this area.)

-I don’t think the average reader will fully appreciate the point at L130 without more context. It’s worth being very explicit about how discordance (as first observed by O’Neill et al. 1992, PNAS) and the timescale of host switching in Drosophila (as calibrated for wRi-like spread by Turelli et al. 2018, CB) help determine what is plausible here. Closely related wMel-like variants have also been observed in several divergent hosts (Martinez et al. 2021, MBE), which for wMel-like Wolbachia in Drosophila is again due to horizontal transfer (plus introgressive transfer) on short timescales (Cooper et al. 2019, Genetics). As the authors note, “we cannot disregard the potential for historic infections that may have driven positive selection at bam”, but I would suggest that we should also consider that the contemporary infections we sample in any given host tend to have been acquired on VERY short evolutionary timescales (thousands of years). I think this is another reason why the authors may not see congruence. This context may be better suited for the L588 paragraph in the discussion if the authors decide to add it.

Original comment - Wolbachia variant and titer: What Wolbachia variant is being used here? A random wMel variant? wMel CS? Is it a sequenced variant? More importantly, do the authors have any data on Wolbachia abundance in host tissues? Almost all wb effects on hosts depend on abundance/localization, and that is probably true here. The question is how much it varies among infected genotypes. Considering titer is likely to improve the interpretation of the results here, but I fully realize these data may not exist, and if so, OK.

Response from authors: This is a good point. As the Wolbachia field has grown, it is important

to note the variants. We have clarified that we are using wMel throughout the paper. We have

not quantified the Wolbachia abundance in the tissue, though we have thought about it. I expect

we would see a correlation between titer and amount of fertility rescue, as you suggest. The

interpretation remains a little tricky as more wildtype-like tissue (more rescue) gives an

opportunity for greater Wolbachia abundance (e.g., there is typically a high density of Wolbachia

in egg chambers, so mutants wish fewer egg chambers will likely have less Wolbachia by nature

of the tissue composition).

At present, we chose to prioritize an experimental suggestion from Reviewer 1, but appreciate

this feedback and will keep it in mind for future experiments. Thank you!

-Understood. I would encourage the authors to say a bit more about titer than they currently do at Line 539 since titer is likely specifically relevant here, in addition to its relevance to most other Wolbachia-host interactions/phenotypes. I think titer data is the one thing that could really improve this paper, although I do not think it is required. A more detailed discussion would be useful.

Original comment - Line 57: It is worth noting that CI reduces egg hatch, but the strength of this reduction can vary widely (i.e., many progeny remain viable depending on the system/context; see Shropshire’s work).

Response from authors: We added the information so our text is not misleading (L61, L64-65).

Thank you.

-The comment at Line 65 is incorrect. Shropshire has recently shown very strong wMel CI by finely tracking CI strength with male age. Others have shown that wMel tends to generally cause strong CI in Ae. aegypti and in D. simulans backgrounds. I suspect D. melanogaster carries a nuclear suppressor of some sort, and that pulling wMel out of its “natural” host and placing it in a novel host that has not evolved suppression results in very strong CI (that can also depend on things like temperature). Theory suggests such suppressors should evolve. Anyway, another sentence and citation of focal work would make this clear to readers.

-None of my suggestions here are necessary/critical except for improving these few areas of scholarship. I really enjoyed learning more about bam, and I think the readers of the journal will appreciate this work.

**Have all data underlying the figures and results presented in the manuscript been provided?**

Reviewer #1: Yes

Reviewer #2: Yes

PLOS authors have the option to publish the peer review history of their article (what does this mean?). If published, this will include your full peer review and any attached files.

Reviewer #1: No

Reviewer #2: No

**Data Deposition**

http://datadryad.org/submit?journalID=pgenetics&manu=PGENETICS-D-23-00319R1

**Press Queries**

---

## [Editor Report · Acceptance letter]

18 Oct 2023

PGENETICS-D-23-00319R1 

Wolbachia infection at least partially rescues the fertility and ovary defects of several new Drosophila melanogaster bag of marbles protein-coding mutants 

Dear Dr Wenzel, 

We are pleased to inform you that your manuscript entitled "Wolbachia infection at least partially rescues the fertility and ovary defects of several new Drosophila melanogaster bag of marbles protein-coding mutants" has been formally accepted for publication in PLOS Genetics! Your manuscript is now with our production department and you will be notified of the publication date in due course.

With kind regards,

Anita Estes

PLOS Genetics

On behalf of:
